# Vasopressin Receptor Type-2 Mediated Signaling in Renal Cell Carcinoma Stimulates Stromal Fibroblast Activation

**DOI:** 10.3390/ijms23147601

**Published:** 2022-07-09

**Authors:** Abeda Jamadar, Nidhi Dwivedi, Sijo Mathew, James P. Calvet, Sufi M. Thomas, Reena Rao

**Affiliations:** 1Jared Grantham Kidney Institute, University of Kansas Medical Center, Kansas City, KS 66160, USA; ajamadar@kumc.edu (A.J.); dwivedi.nid25@gmail.com (N.D.); jcalvet@kumc.edu (J.P.C.); 2Department of Pharmaceutical Sciences, School of Pharmacy, North Dakota State University, Fargo, ND 58105, USA; sijo.mathew@ndsu.edu; 3Department of Biochemistry and Molecular Biology, University of Kansas Medical Center, Kansas City, KS 66160, USA; 4Department of Cancer Biology, University of Kansas Medical Center, Kansas City, KS 66160, USA; 5Department of Otolaryngology, University of Kansas Medical Center, Kansas City, KS 66160, USA; sthomas7@kumc.edu; 6Department of Medicine, 5040 WHE, The Jared Grantham Kidney Institute, University of Kansas Medical Center, 3901 Rainbow Blvd, Kansas City, KS 66160, USA

**Keywords:** cancer-associated fibroblasts, vasopressin type-2 receptor, clear cell renal cell carcinoma, myofibroblasts, OPC31260, dDAVP, yes associated protein

## Abstract

Vasopressin type-2 receptor (V2R) is ectopically expressed and plays a pathogenic role in clear cell renal cell carcinoma (ccRCC) tumor cells. Here we examined how V2R signaling within human ccRCC tumor cells (Caki1 cells) stimulates stromal cancer-associated fibroblasts (CAFs). We found that cell culture conditioned media from Caki1 cells increased activation, migration, and proliferation of fibroblasts in vitro, which was inhibited by V2R gene silencing in Caki1 cells. Analysis of the conditioned media and mRNA of the V2R gene silenced and control Caki1 cells showed that V2R regulates the production of CAF-activating factors. Some of these factors were also found to be regulated by YAP in these Caki1 cells. YAP expression colocalized and correlated with V2R expression in ccRCC tumor tissue. V2R gene silencing or V2R antagonist significantly reduced YAP in Caki1 cells. Moreover, the V2R antagonist reduced YAP expression and myofibroblasts in mouse xenograft tumors. These results suggest that V2R plays an important role in secreting pro-fibrotic factors that stimulate fibroblast activation by a YAP-dependent mechanism in ccRCC tumors. Our results demonstrate a novel role for the V2R-YAP axis in the regulation of myofibroblasts in ccRCC and a potential therapeutic target.

## 1. Introduction

CcRCC is the most common cancer of the kidneys, causing ~15,000 deaths per year in the United States [1]. It is highly invasive, metastatic, and often resistant to radiation and chemotherapy [2]. Cancer-associated fibroblasts (CAFs) are known to regulate certain hallmarks of cancer, including proliferation, angiogenesis, invasion, and metastasis, and the extent of their presence often correlates with poor clinical outcomes in human tumors [3,4,5,6,7,8,9]. Among other markers, CAFs express αSMA, a distinctive feature of fibroblast to myofibroblast activation that provides contractility to these cells [10]. ccRCC tumor cells are known to undergo EMT to transform into a myofibroblast-like phenotype which helps in metastasis, and myofibroblast-related genes are upregulated in metastatic tissue from ccRCC patients [6,11]. CAFs also play a major role in drug resistance in ccRCC via a TDO/Kyn/AhR-dependent signaling pathway [12]. Recent studies have associated the presence of CAFs with adverse prognostic effects as well as the induction of tumor resistance to thymidine kinase inhibitors in ccRCC [13]. Reciprocal interactions exist between tumor cells and CAFs whereby tumor cells stimulate CAF activation, while CAFs, in turn, regulate tumor growth, angiogenesis, and metastasis [14]. The pathogenicity of CAFs is attributed to their ability to secrete cytokines and growth factors, regulate other immune cells, reprogram tumor cell metabolism, excessively secrete extracellular matrix, and remodel tissue [11].

Vasopressin is a neuropeptide hormone that plays an essential role as an antidiuretic hormone via the V2R, vasoconstrictor via its V1a receptors, and stimulator of the release of ACTH by pituitary corticotropes via its V1b receptors. In the current study, we investigated whether V2R signaling within ccRCC tumor cells regulates CAFs by paracrine mechanisms. AVP binding to the V2R, a G-protein coupled receptor, triggers a signaling cascade involving increased adenylate cyclase activity, intracellular cAMP levels, and protein kinase-A (PKA) activity leading to water reabsorption and maintenance of water homeostasis by the kidneys [15]. However, V2R activity is known to promote cyst expansion in polycystic kidney disease (PKD) by stimulating cystic epithelial cell proliferation and cyst-filling fluid secretion [16,17]. V2R is normally expressed within the collecting duct, connecting tubule and thick ascending limb segments of the renal nephron, but not the proximal tubules [15]. ccRCC tumors are known to originate from the proximal tubules [18,19,20,21]. We recently found that V2R is ectopically expressed in ccRCC tumor cells [22]. Moreover, V2R signaling via the cAMP-PKA-ERK1/2/MAPK pathway drives cell proliferation of ccRCC tumor cells [22]. Moreover, V2R agonists increased tumor growth, while V2R antagonists, including Tolvaptan, an FDA-approved drug for hyponatremia and PKD, suppressed tumor growth [22]. In the current study, we show that V2R agonist treatment can increase CAFs in mouse xenograft tumors, while V2R antagonist can reduce their numbers. In vitro studies show that V2R activity, through a YAP-dependent mechanism, regulates the secretion of fibroblast activating factors by ccRCC tumor cells, demonstrating a new mechanism by which ccRCC tumor cells regulate fibroblasts in their microenvironment.

## 2. Results

### 2.1. Gene Silencing of V2R in ccRCC Tumor Cells Reduces Fibroblast Activation, Proliferation, and Migration

In a recent study, we found that ccRCC tumor growth was significantly increased by treatment with dDAVP, a V2R agonist, while OPC31260, a V2R antagonist, reduced tumor growth in a mouse xenograft model [22]. Now upon further examination of these xenograft tumor tissues, we observe a significant increase in the myofibroblast population as suggested by αSMA expression in the dDAVP treatment group and reduced αSMA expression in the OPC31260 treatment group when compared to the vehicle treatment group (Figure 1A,B). This finding suggests that V2R activity could regulate CAFs. Examination of human ccRCC tumors shows V2R expression in tumor cells only but not in the αSMA expressing CAFs (Figure 1C). To determine the role of this ccRCC tumor cell-specific V2R in the regulation of fibroblasts, undifferentiated rat renal fibroblasts (NRK-49F cells) were exposed to serum-free cell culture conditioned media (CM) from Caki1 cells, a human ccRCC tumor cell line. Exposure to Caki1-CM increased the viability and proliferation of these NRK-49F cells (Figure 1D,E). To further determine the role of V2R on paracrine fibroblast activation, we exposed NRK-49F cells to CM from Caki1 cells transfected with scrambled (Scr) or V2R-SiRNA (V2R-Si) to knockdown V2R expression (Appendix A. Exposure of NRK-49F cells to Scr-CM significantly increased αSMA protein levels when compared to cells exposed to normal control media (Figure 1F,G), suggesting fibroblast to myofibroblast differentiation. In comparison, NRK-49F cells exposed to V2R-Si-CM showed significantly reduced αSMA levels (Figure 1F,G), cell viability (Figure 1H), and migration (Figure 1I and Appendix A) when compared to Scr-CM. These results suggest that V2R regulates the secretion of ccRCC tumor cell-mediated factors that regulate renal fibroblast activation, proliferation, and migration.

### 2.2. V2R Regulates Secreted Factors Produced by Caki1 Human ccRCC Cell Line

To identify the V2R-regulated factors secreted by ccRCC tumor cells, gene array and protein array analyses were performed. Gene array analysis for fibrosis and inflammation-related secreted factors, including ECM components, proteinases, chemokines, growth factors, and matricellular proteins, was performed in Scr and V2R-SiRNA transfected Caki1 cells. The V2R-Si group showed significantly reduced mRNA levels for CSF2, PAI-1, AREG, TFPI-2, IL8, CCL2, CCL5, and TSP1, and significantly increased levels for MMP1, TIMP1, ICAM1, IL1β, CXCL10, and CYR61, as compared to the Scr group (Figure 2A). We next analyzed secretory proteins from the CM of V2R-Si and Scr Caki1 cells using a commercially available cytokine protein array. Multiple factors could be detected in the CM, of which IL6, IL8, CCL2, CCL5, CCL20, GM-CSF, IGFBP-a, MIF, UPAR, and TNFR1 were reduced in the V2R-Si-CM group (Figure 2B,C). Taken together, both the mRNA and protein levels of IL8 (CXCL8 gene), CCL2, CCL5, and GM-CSF (CSF2 gene) were reduced in the V2R-Si-CM group when compared to the Scr-CM group suggesting that these secreted factors are regulated by V2R. Furthermore, we found increased mRNA levels of IL8, CCL2, CCL5, and CSF-2 in human ccRCC tumors compared to non-malignant kidney tissues (Figure 2D). mRNA levels of other factors such as TFPI-2, TSP1, AREG, and PAI-1, which were found to be regulated by V2R gene silencing in Caki1 cells (Figure 2A), were also increased in human ccRCC tumors compared to non-malignant kidney tissues (Figure 2D). Analysis of The Cancer Genome Atlas (TCGA) kidney clear cell carcinoma (KIRC) database for RNA sequencing data related to ccRCC showed significantly reduced overall survival in high expressers of CXCL8 (IL8), CCL5, CSF2, TFPI-2, and SERPINE1 (PAI1) (Appendix A).

### 2.3. V2R Regulates YAP in ccRCC Tumor Cells

Since multiple secreted factors identified above, including AREG, PAI-1, CCL2, CCL5, IL6, IL8, and CSF2, are also known to be regulated by yes associated protein (YAP), we investigated whether V2R regulates YAP in ccRCC tumor cells. In human ccRCC tumor tissue, nuclear YAP expression was detected in V2R expressing ccRCC tumor cells (Figure 3A), and YAP mRNA was detected in ccRCC human tissues when compared with non-malignant kidney tissues (Figure 3B). YAP expression was also found to correlate with V2R expression in ccRCC tumor samples using the TCGA KIRC database (Figure 3C) positively. YAP expression also correlates with CCL5, CSF2, TSP1, CCL2, and CXCL8 (Figure 3D–H).

To determine if YAP regulates any of the V2R-regulated secreted factors identified in Figure 2, we treated Caki1 cells with verteporfin. Verteporfin reduces YAP transcriptional activity by inhibiting the YAP-TEAD interaction and increasing 14-3-3σ, which sequesters YAP in the cytoplasm [23,24]. Verteporfin treatment significantly reduced PAI-1, CCL2, CCL5, TSP1, and TFPI-1 mRNA levels in Caki1 cells (Figure 4A). Caki1 cells showed high YAP protein expression, which was not significantly further increased by dDAVP treatment (Figure 4B,C). However, the V2R antagonist OPC31260 treatment significantly reduced YAP expression (Figure 4B,C). These doses of verteporfin, dDAVP, and OPC have been used in our prior studies [22,25]. Gene silencing of V2R in Caki1 cells also reduced YAP expression in both the vehicle-treated as well as dDAVP-treated cells (Figure 4D,E). These results show that V2R gene silencing and V2R antagonist treatment can reduce YAP levels in Caki1 cells, thereby suggesting that V2R regulates YAP in ccRCC tumor cells. Consistent with this result, in the mouse xenograft tumor tissue from the study described in Figure 1A, YAP expression was found in tumors of vehicle-treated mice, in comparison to which YAP was significantly reduced in the OPC31260 treatment group. However, no significant change was observed in the dDAVP treatment group as compared to the control group (Figure 4F,G).

## 3. Discussion

This study shows a novel mechanism by which ccRCC tumor cells regulate stromal fibroblasts through a V2R-YAP mediated mechanism. We show for the first time that (A) V2R activity in ccRCC tumor cells is important for fibroblast activation, migration, and proliferation, (B) V2R regulates the production of secreted factors from ccRCC tumor cells that are known to activate CAFs, and (C) secretion of the factors, including CCL2, CCL5, GM-CSF, IL8, TSP1, and TFPI-1 are regulated by a V2R and YAP -dependent mechanism in ccRCC cells.

Reciprocal interactions between tumor cells and CAFs occur in a variety of different cancers [26,27]. For instance, human ccRCC tumor cell lines are known to induce periostin accumulation and to activate NIH3T3 mouse fibroblasts, and these activated fibroblasts then enhance ccRCC cell attachment in vitro [9]. Similarly, in ovarian cancers, the presence of stromal myofibroblasts enables the invasion of endothelial cells and correlates with the exit of tumor cells from dormancy [28,29]. Here we show that V2R activity within ccRCC tumor cells regulates CAFs in the tumor microenvironment by a paracrine mechanism. Our study demonstrates that tumor cells of epithelial cell origin can regulate myofibroblasts by a V2R-dependent mechanism. We found dense populations of αSMA-expressing CAFs around V2R-expressing tumor cells in human ccRCC tumors. The V2R selective agonist dDAVP increased CAFs in ccRCC mouse xenograft tumors, while a V2R antagonist significantly reduced CAFs in ccRCC tumors. Importantly, conditioned culture media from Caki1 cells stimulated activation, proliferation, and migration of fibroblasts in vitro, which was significantly reduced using conditioned media from Caki1 cells following gene silencing of V2R.

This study reveals a novel mechanism by which ccRCC tumor cells can regulate stromal fibroblasts through V2R-YAP mediated secretion of fibroblast activating factors. YAP and its homolog TAZ (PDZ binding motif) are transcriptional co-activators of TEAD, which regulate cell proliferation, differentiation, and apoptosis [30]. Activation of the Hippo signaling cascade leads to phosphorylation, nuclear exclusion, cytoplasmic sequestration, and proteolytic degradation of YAP and TAZ [31,32,33]. ECM stiffness mechano-activates YAP/TAZ within fibroblasts as well as epithelial cells and promotes the production of profibrotic mediators and ECM proteins by fibroblasts and proliferation and survival of epithelial cells [34]. Dysfunctional Hippo signaling in both tumor cells, as well as myofibroblasts, is known to regulate cell proliferation, migration, invasion, angiogenesis, and drug resistance in ccRCC [35,36,37,38]. However, the role of tumor cell-specific YAP in the paracrine regulation of CAFs in ccRCC has not been previously examined. Our study indicated that V2R regulates YAP in ccRCC tumors as a V2R antagonist and V2R gene silencing reduced YAP in vitro and in vivo. Previously, we reported that in PKD kidneys, V2R regulates myofibroblast activation and disease progression by a YAP and connective tissue growth factor (CCN2)-dependent mechanism [25].

The current study also provides evidence that YAP plays an important role in the secretion of V2R-mediated fibroblast-regulating factors. Our finding that V2R in tumor cells can regulate CAFs in ccRCC tumors suggests that secreted factors are involved in mediating this interaction. Our analysis of ccRCC tumor cell mRNA, as well as proteins in the conditioned media of these cells, demonstrates for the first time that V2R-dependent signaling can regulate multiple pro-inflammatory and pro-fibrotic secreted factors, including cytokines, chemokines, growth factors, proteinases, and matricellular proteins. V2R gene silencing, as well as YAP inhibition, reduced multiple known YAP-regulated factors. The observation that V2R and YAP can regulate multiple pro-inflammatory factors suggests that V2R and YAP could possibly also regulate macrophages, T-cells, and granulocytes in ccRCC tumor development. A limitation of this paper is that the experiments were performed in the Caki1 cell line, which is a VHL wildtype cell line, while 70–80% of ccRCC patients have mutations in VHL. Our earlier study showed that 786-0 cells (VHL mutant human RCC cell line) express V2R [22]. We also showed that V2R antagonist OPC31260 could inhibit cell viability, colony formation and migration, and cause cell cycle arrest of 786-0 cells [22].

In conclusion, these findings provide new insights into a novel pathogenic mechanism by which V2R, an epithelial-specific hormone receptor, regulates CAFs in ccRCC tumors by a YAP and secreted factor-mediated mechanism. Pharmacological approaches targeting the V2R-YAP molecular axis may have strong implications for therapy for ccRCC.

## 4. Materials and Methods

### 4.1. ccRCC Subcutaneous Mouse Xenograft Studies

Athymic Nude-Foxn1^nu^ mice (Nu/Nu mice from Envigo/Harlan), female, 7–8 weeks old, and weighing ~25 g were subcutaneously injected with 1 × 10^6^ Caki1 cells in 100 µL of DMEM medium on the right flank. When palpable tumors appeared, tumor volumes were measured using calipers (Tumor volume = (length × width^2^)/2). When the tumor volumes reached ~80–100 mm^3^, mice were randomized and assigned to 3 groups and administered vehicle (saline), OPC31260 (30 mg/Kg BWt), or dDAVP (1 μg/Kg BWt, IP, daily for 28 days) purchased from Sigma Millipore (St. Louis, MO, USA). Body weights and tumor volumes were measured every other day. At sacrifice, tumors were collected, weighed, and fixed in 4% paraformaldehyde for immunostaining. All animal studies were performed according to the protocols approved by the University of Kansas Medical Center Institutional Animal Care and Use Committee.

### 4.2. Human Tissues and Cells

Human ccRCC tumor tissue samples were obtained from the KUMC Cancer Center’s Biorepository Core. Caki1 ccRCC cell line (ATCC^®^ HTB-46™) and NRK49F rat renal fibroblasts (ATCC^®^ CRL-1570) cell lines were used.

### 4.3. Western Blot

Cell lysate in SDS Laemmli buffer were loaded onto 10% SDS-polyacrylamide agarose electrophoresis gels as described before [39,40]. Primary antibodies for YAP and GAPDH (Santa Cruz Biotechnology, Inc., Dallas, TX, USA) and αSMA (Abcam) and V2R (#V5514) from Sigma Millipore (St. Louis, MO, USA) were used. Secondary antibodies were purchased from Agilent Dako (Santa Clara, CA, USA) and ECL reagent (PerkinElmer, Netherlands). 4 × 10^5^ Caki-cells were plated in a 6 well plate. The cells were grown in complete media containing 10% FBS and 1% penicillin-streptomycin. When the cells reached 60–70% confluency, the complete media was replaced with serum-free media overnight. After aspirating the serum-free media, the cells were then treated with 1 nM dDAVP or 25 μM OPC or control medial for 24 h in 0.2% FBS and 1% penicillin-streptomycin media. The cells were lysed using SDS Laemmli buffer and analyzed for western blot.

### 4.4. Immunohistochemistry/Immunofluorescence (IHC/IF)

Fixed and paraffin tissue sections were processed as described before [41]. The following primary antibodies were used: αSMA (Abcam, Cambridge, MA, USA), YAP (Santa Cruz Biotechnology, Dallas, TX, USA), V2R (Millipore Sigma, St. Louis, MO, USA), BrdU (Cell Signaling Technology, Danvers, MA, USA), and V2R (#V5514) from Sigma Millipore (St. Louis, MO, USA). For IHC, secondary antibodies were applied, followed by incubation with Streptavidin HRP conjugate (Invitrogen, New York, NY, USA) and slides were developed with DAB (Vector Laboratories, Burlingame, CA, USA) and counterstained with Harris Haematoxylin, dehydrated, and mounted with Permount (Fisher Scientific, Fair Lawn, NJ, USA). For IF, goat anti-Rabbit IgG fluor and Goat anti-mouse IgG Texas red (Invitrogen, New York, NY, USA), secondary antibodies were applied, incubated, washed with PBST, and stained with DAPI. Slides were mounted with Flour-G (Invitrogen, New York, NY, USA) and sealed with nail polish. All images were captured using a Nikon 80i upright microscope (Tokyo, Japan) in the KUMC Imaging Center.

### 4.5. Quantitative Real-Time PCR

RNA was isolated using the trizol method (Ambion, Austin, TX, USA). A high-capacity CDNA reverse transcription kit from Applied Biosystems (Foster City, CA, USA) was used to make cDNA according to the manufacturer’s protocol. 4 × 10^5^ Caki- cells were plated in a 6 well plate. The cells were grown in complete media containing 10% FBS and 1% penicillin-streptomycin. When the cells reached 60–70% confluency, the complete media was replaced with serum-free media overnight. After aspirating the serum-free media, the cells were then treated with 2.5 μM verteporfin purchased from Sigma Millipore (St. Louis, MO, USA). or control media for 16 h in 0.2% FBS and 1% penicillin-streptomycin media. The cells were lysed in trizol and QRTPCR was performed using power SYBR Green PCR master mix Applied Biosystems (Foster City, CA, USA) following the manufacturer’s protocol.

The PCR for human cDNA samples was carried out using EmeraldAmp MAX PCR Master Mix (Takara Bio, Bath, UK). The primer list is provided in Appendix A.

### 4.6. Cell Culture Conditioned Media Collection

Caki1 cells were grown in McCoy’s 5A Medium (ATCC^®^ 30-2007™) containing 10% fetal bovine serum (FBS) and 1% Penicillin-streptomycin in 100 mm plates. When fully confluent, the media was replaced with serum-free media. Conditioned media was collected after 24 h, centrifuged to remove debris, and used directly for studies. Caki1 cells used in these studies were in their 4th to 10th passages.

### 4.7. Fibroblast to Myofibroblast Differentiation

NRK-49F cells were grown in 100mm plates in high glucose DMEM media (ATCC^®^ 30-2002™) containing 5% FBS and 1% Penicillin-streptomycin (Pen/Strep). When 50% confluent, NRK-49F cells were exposed to serum-free conditioned media from Caki1 cells for 48 h. Conditioned media was changed every 24 h. Cells were lysed and αSMA levels were measured by immunoblotting.

### 4.8. Migration Assays

NRK-49F were seeded in 6-well plates and grown until confluent in 5% FBS and 1% Pen/Strep containing DMEM medium. When confluent, the NRK-49 F cells were replaced with either control serum-free media (vehicle) or serum-free conditioned media collected from Caki1 cells (as described above). A sterile 200 μL pipet tip was used to place a scratch (wound) in the cell monolayer, followed by washing with PBS to remove dislodged cells. The wounds were then photographed (2× magnification) at different time points and the wound closure was measured using ImageJ. A separate set of plates with a similar treatment was used to assess cell proliferation to adjust % wound closure to cell proliferation.

### 4.9. Cell Viability and Cell Proliferation Analysis

For MTT assay [42], 25,000 NRK49F cells were seeded in 24-well plates and grown in 5% FBS and 1% Pen/Strep containing DMEM medium. When cells were 40–50% confluent, they were washed with PBS and media was replaced with serum-free conditioned media collected from Caki1 cells or control serum-free media (vehicle) for 48 h. Following this, cells were incubated in 5 mg/mL MTT solution for 2 h, following which the intracellular purple formazan was solubilized in DMSO and quantified by spectrophotometry at 540 nM.

For BrdU incorporation assay [43], cells were seeded on coverslips in 24-well plates and incubated in serum-free conditioned media collected from Caki1 cells or control serum-free media (vehicle) for 48 h as described above. In the last 3 h of treatment, all cells were also treated with BrdU (3 µg/mL). Cells were fixed and immunostained for DAPI and BrdU, imaged, quantified, and expressed as the ratio of total BrdU positive to total DAPI stained nuclei.

### 4.10. V2R Gene Silencing

V2R-Si (AVPR2-SiRNA, 4390824; Clone 1: S1842, Clone 2: S1843) from Ambion were used to knockdown V2R in cells. Scr (scrambled siRNA, AM4620) was used as a negative control [22]. Cells were plated in a six-well plate at a density of 2.5 × 10^5^ cells/well and grown in 10% FBS and 1% Pen/Strep containing McCoy’s medium. When the cells reached 30–40% confluency, they were transfected with scrambled or V2R SiRNA clones with a final concentration of 5 nM each in a 1:1 ratio or control Scr (10 nM) using Lipofectamine^®^ RNAiMAX (Invitrogen, Carlsbad, CA, USA) following the manufacturer’s protocol. After 4 h of transfection, media was replaced with fresh complete media containing 10% FBS and 1% Penicillin-streptomycin and left for 48 h in the incubator. Cells were used for different experiments. Knockdown efficiency of the siRNAs was evaluated by the Western blot method.

### 4.11. Cytokine Protein Array for Conditioned Media

Secretory factors were determined using a cytokine array (Human Cytokine Antibody Array C Series 6 and 7 1000) procured from RayBiotech, Inc. Peachtree Corners, GA, USA. Sub-confluent cultures of Caki1 cells transfected with Scr and V2R-Si were incubated in serum-free media for 48 h, and conditioned media (CM) was collected. The antibody pre-coated membranes were incubated in 48 h CM and all the steps were followed as per the manufacturer’s protocol. Chemiluminescence was detected using an Amersham Imager and densitometry of the signal was measured using ImageJ software. Sample values in each membrane were subtracted from blank values and normalized to their positive control.

### 4.12. Statistics

Values are expressed as mean ± standard error for all in vivo studies and mean ± standard deviation for in vitro studies. Data were analyzed by a two-tailed unpaired t-test with Welch’s correction using GraphPad Prism software (Version 9). A probability level of 0.05 (*p* ≤ 0.05) was considered significant.

## Figures and Tables

**Figure 1 ijms-23-07601-f001:**
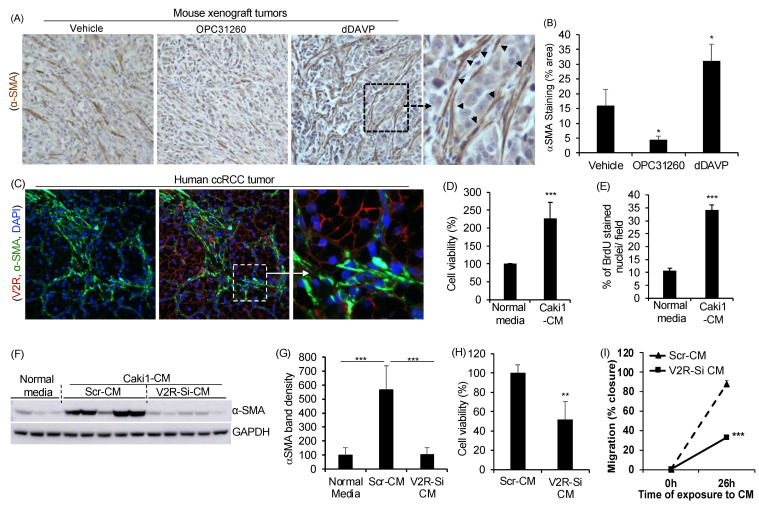
V2R in ccRCC tumor cells is important for paracrine regulation of fibroblasts: (**A**) Nu/Nu mice were inoculated subcutaneously with Caki1 cells. Ten days later, mice were treated with vehicle OPC31260 (30 mg/Kg) or dDAVP (1 μg/Kg) daily for 28 days. Immunostaining of tumor tissue sections for αSMA (brown). (**B**) Quantitation of immunostaining. (*n* = 4 tumors per group). (**C**) In human ccRCC tumor tissue, immunostaining for αSMA (green), V2R (red), and DAPI (blue). (**D**) Cell viability measured by MTT assay on NRK-49F renal fibroblasts exposed to serum-free conditioned cell culture media (CM) from Caki1 cells for 48 h. (*n* = 6), and (**E**) Cell proliferation of NRK-49F cells treated with Caki1-CM for 48 h measured as % BrdU positive nuclei/total DAPI stained nuclei/field/experiment (*n* = 5). Cells were exposed to BrdU (3 µg/mL) for 3 h before the end of the study. (**F**) Immunoblot of NRK-49F cells exposed to normal control media (media not previously exposed to any cells) or CM from Caki1 cells transfected with V2R-SiRNA (V2R-SiRNA-CM) or Scr (Scr-CM) for 48 h. and (**G**) quantitation of band density relative to GAPDH. (**H**) MTT assay on NRK-49F cells exposed to Scr-CM or V2R-SiRNA-CM for 48 h (*n* = 6) and (**I**) % wound closure in the scratch assay of NRK-49F cells exposed to Scr-CM or V2R-SiRNA-CM (*n* = 5). * *p* < 0.05, ** *p* < 0.01, *** *p* < 0.001 by *t*-test.

**Figure 2 ijms-23-07601-f002:**
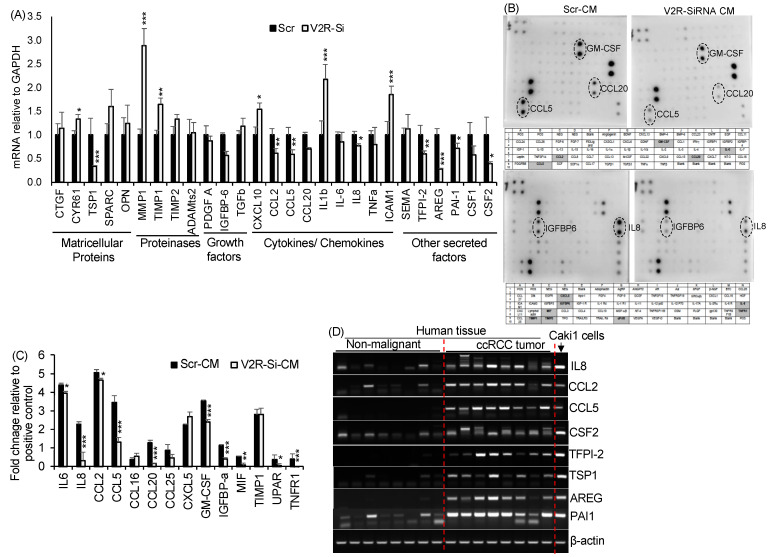
V2R regulates secreted factors produced by ccRCC tumor cells: (**A**) QRTPCR for mRNA levels of secreted factors relative to GAPDH in Caki1 cells transfected with Scr or V2R-SiRNA for 48 h. *n* = 4. (**B**) Antibody arrays show proteins in Scr-CM or V2R-SiRNA-CM from Caki1 cells. Each antibody is spotted in duplicate vertically. (**C**) Quantitation of band density relative to positive control based on antibody array shown in B. (**D**) mRNA measured by RTPCR. Each band represents one human ccRCC tumor or non-malignant kidney tissue sample. * *p* < 0.05, ** *p* < 0.01, *** *p* < 0.001 vs. Scr by *t*-test.

**Figure 3 ijms-23-07601-f003:**
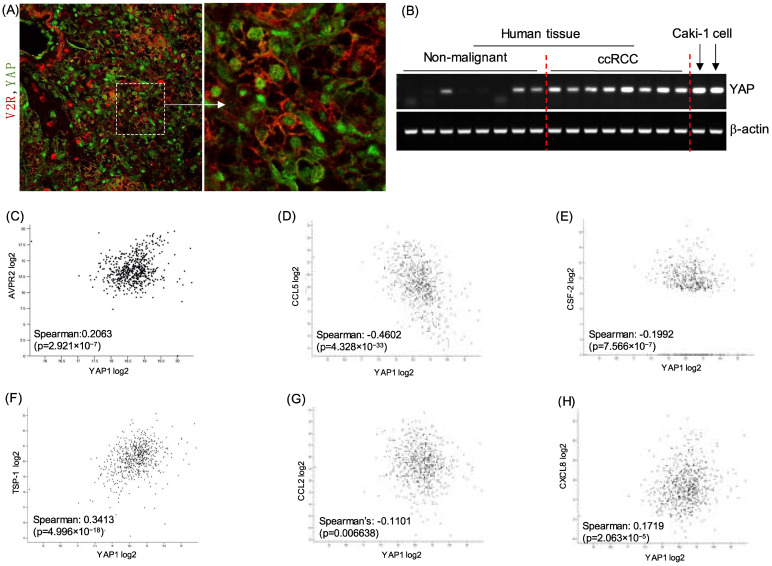
YAP expression in ccRCC tumor cells and correlation with V2R regulated secreted factors: (**A**) Immunostaining for V2R (red) and YAP (green) in human ccRCC tumor tissue, (**B**) YAP mRNA levels in human ccRCC tumor and non-malignant kidney tissue. (**C**) Correlation plot using GDC TCGA Kidney Clear Cell Carcinoma (KIRC) Cohort (*n* = 985) for YAP1 vs. V2R (AVPR2 gene), (**D**) YAP1 vs. CCL5, (**E**) YAP1 vs. CSF2, (**F**) YAP1 vs. TSP1, (**G**) YAP1 vs. CCL2, (**H**) YAP1 vs. CXCL8. Gene expression RNAseq-HTSeq-FPKM-UQ Unit: log2 (fpkm-uq + 1) shown in C to H.

**Figure 4 ijms-23-07601-f004:**
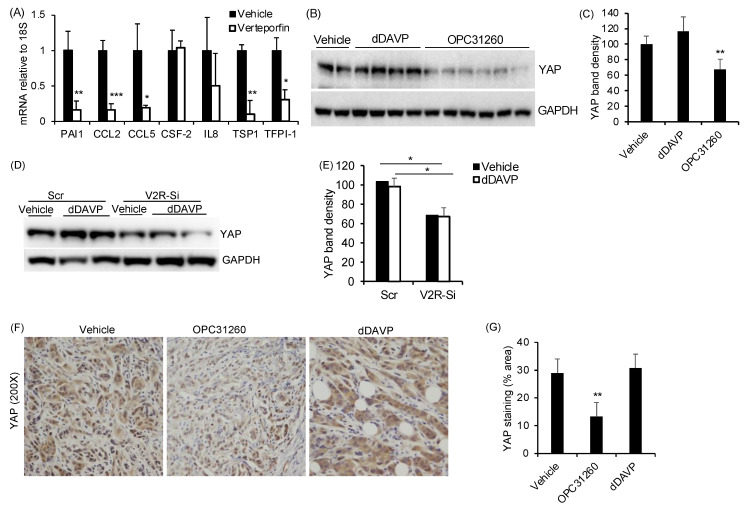
V2R regulates YAP in ccRCC tumor cells: (**A**) mRNA levels of secreted proteins relative to 18 S mRNA in Caki1 cells treated with YAP inhibitor, verteporfin (2.5 µM for 16 h) (*n* = 4). (**B**) Immunoblot for Caki1 cells treated with vehicle, dDAVP (1nM) or OPC31260 (25 µM) for 24 h, and (**C**) Quantitation of band density. (**D**) Immunoblot for YAP in Caki1 cells transfected with Scr or V2R-SiRNA for 48 h and treated with vehicle or dDAVP (1 nM, 30 min), and (**E**) Quantitation of band density. (**F**) Immunostaining of mouse xenograft tumor tissue sections for YAP (brown) from study shown in Figure 1A. (**G**) Quantitation of staining intensity for YAP in mouse tumor. (*n* = 4 tumors per group). * *p* < 0.05, ** *p* < 0.01, *** *p* < 0.001 vs. vehicle by *t*-test.

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
