# Peer review of "Vasopressin Receptor Type-2 Mediated Signaling in Renal Cell Carcinoma Stimulates Stromal Fibroblast Activation"

_ijms, 2022, doi:10.3390/ijms23147601_

Round 1
Reviewer 1 Report
In this manuscript Jamadar et al. studied stimulation of stromal cancer associated fibroblasts by vasopressin type-2 receptor (V2R) activation in human renal cell carcinoma tumor cells (Caki-1 cells). The action of cell culture conditioned media obtained from Caki-1 cells was determined to increase migration and proliferation of fibroblasts. This effect was inhibited by silencing of the V2R gene in Caki-1 cells. V2R action in Caki-1 cells was found to be dependent on YAP mechanisms. This manuscript addresses an important question; it is well designed and well written. Specific points:
Results
Please show (or report relevant references on) knockdown of V2R expression by V2R-SiRNA (V2R-Si) in Caki-1 cells.
Please provide the rational (or report relevant references) for selecting specific treatment concentrations and times [dDAVP (1nM) or OPC31260 (25µM) for 24h; verteporfin (2.5 µM for 16h)].
Figure 2. It is confusing as presented the labeling of panel A and C conditions. In the panel A, V2R-Scr is indicated in black, and it is in white in panel C. Please keep the labeling consistent.
Methods
Page 8, lines 277 and 278 “Caki-1 cells, were grown in McCoy's 5A 277 Medium (ATCC® 30-2007™) in 100mm plates in DMEM media (Catalog # SH30023.01, 278 HyClone)”. Please clarify this sentence and provide more information about the cell culture conditions of Caki-1 cells (e.g., passage numbers, etc.).
Page 1, line 24, please correct “corelated” with “correlated”.
Author Response
File is attached

Reviewer 2 Report
In this article, Hamadar A. et al. want to show how V2R through YAP modulate different secreted factors that acts paracrine in CAFs. From my point of view, there are some points to improve in this article before to be published.
Major
- All experiments performed are in caki-1 cell line, which is a VHL WT cell line, but most of the ccRCC patients, around 70-80%, have mutations in VHL. Authors should do the experiments in more than one cell line and preferably in cell lines representative of the patients.
- Authors do not directly demonstrate that secreted factors due to V2R or YAP modulation are responsible for a change in fibroblasts. They only show that V2R modulation has effects on fibroblasts, in terms of a-SMA expression, viability and migration. Furthermore, the authors have not looked at the contractility of these fibroblasts and other processes.
- Line 87-88 and Figure 1: Caki-1 cells transfected with V2R-scrambled (V2R-Scr) or V2R-SiRNA (V2R-Si) to knockdown V2R expression. Authors should show the knockdown in V2R expression when they used a V2R-siRNA.
- Figure 2D: Why did the authors select these secreted factors to validate in human tissue? Some of them are found in both gene array and protein array, but only some of the others were validated. Moreover, in the figure, the authors add the Caki-1 cell line, but it would be interesting to add Caki-1 V2R-scr and Caki-1 V2R-siRNA.
- Figure 2D: Authors should add V2R levels in human tissues
- Figure 2D and 3B: The number of patients analyzed are different, in the figure 2D authors showed 16 patients and in the figure 3B authors showed 17 patients, are not the same patients in both cases?
- Figure 3C-3H: Why authors decided to analyze the correlation between YAP expression and some secreted factors by two different two correlation coefficients (Pearson and Spearman)?
- Lines 171-175: Authors said that, in the mice experiment, YAP expression is increased in the dDAVP treated group, but it is not true (figure 4F and 4G), so authors must modify this sentence and the sentence in the discussion section (line 221: while a V2R agonist increased YAP), because authors did not demonstrate it, neither in in vitro experiments (caki-1 cell line) nor in vivo experiments (mice with caki-1 cell line).
- Lines 162-165 and Figure 4A: Why authors used Verteporfin to reduce YAP instead of siRNA? How can they be sure that the effect they observed on secreted factors is a result of YAP reduction and not some other effect caused by the drug? Could also authors show an image to demonstrate de YAP downregulation?
- Figure 4A: Authors show the mRNA levels relative to 18S, but in the Figure 2A they showed mRNA levels relative to GAPDH; authors should use the same reference gene in all cases.
- Figure 4D: Why authors showed only one sample of Veh (I guess vehicle) and two different samples for the treatment with dDAVP.
- Why do the authors use rat renal fibroblast cell line for in vitro experiments instead of human renal fibroblast cell line? The use of human fibroblasts would be more correct.
- Materials and methods section: some information detailed below is missing:
o Antibodies used for western blot and immunohistochemistry/immunofluorescence, the reference and the clone used should be indicated.
o Quantitative real-time PCR is do it by SYBR? Detail it and how many RNA/cDNA is used.
o Authors wrote that Caki-1 cell lines were grown in McCoy’s 5A Medium and also in DMEM media, why two different mediums? Without any additives (eg. antibiotics, FBS, …). And NRK49F, how is it grown?
o Wound closure assays: It remains to detail the objective of the microscope used and how the % of closure has been measured. In addition, the authors said that the cells were grown at 10% FBS, normally for this type of experiment the FBS is removed or reduced, avoiding cell death.
o For siRNA experiments, the concentration of cells for this experiment and the concentration of siRNA used is missing.
o Authors should include in this section an explanation about experiments with the V2R agonist and antagonist in cell lines, such concentration of cells, concentration of agonist/antagonist, time, …
- Supplemental Figure 3: is not mentioned in the main text, in blue is DAPI instead of dDAVP, and this figure need an analysis because in the images the result is not clear.
Minor
- Figure 1F: the line that marks the bands of V2R-Scr-CM and V2R-si-CM are not well defined, please correct it.
- Figure 1G: Authors wrote YAP band density instead of α-SMA band density (Y-axis)
- Figure 2A and 2C: It would be better if in both graphs authors use the same color for Scr and siRNA
Round 2
Reviewer 1 Report
The authors responded adequately to the points raised by this Reviewer. Thank you.